# Environment-Assisted Modulation of Heat Flux in a Bio-Inspired System Based on Collision Model

**DOI:** 10.3390/e24081162

**Published:** 2022-08-20

**Authors:** Ali Pedram, Barış Çakmak, Özgür E. Müstecaplıoğlu

**Affiliations:** 1Department of Physics, Koç University, Sarıyer, Istanbul 34450, Türkiye; 2College of Engineering and Natural Sciences, Bahçeşehir University, Beşiktaş, Istanbul 34353, Türkiye; 3TÜBİTAK Research Institute for Fundamental Sciences, Gebze 41470, Türkiye

**Keywords:** open quantum systems, quantum thermodynamics, collision model

## Abstract

The high energy transfer efficiency of photosynthetic complexes has been a topic of research across many disciplines. Several attempts have been made in order to explain this energy transfer enhancement in terms of quantum mechanical resources such as energetic and vibration coherence and constructive effects of environmental noise. The developments in this line of research have inspired various biomimetic works aiming to use the underlying mechanisms in biological light harvesting complexes for the improvement of synthetic systems. In this article, we explore the effect of an auxiliary hierarchically structured environment interacting with a system on the steady-state heat transport across the system. The cold and hot baths are modeled by a series of identically prepared qubits in their respective thermal states, and we use a collision model to simulate the open quantum dynamics of the system. We investigate the effects of system-environment, inter-environment couplings and coherence of the structured environment on the steady state heat flux and find that such a coupling enhances the energy transfer. Our calculations reveal that there exists a non-monotonic and non-trivial relationship between the steady-state heat flux and the mentioned parameters.

## 1. Introduction

Photosynthesis is the natural photochemical process carried out by certain biological organisms, in which the solar energy is converted into chemical energy and used by the organism itself. Initially, the energy from light is absorbed by the light harvesting antenna complex (a network of chromophores), and the excitation energy is transferred to the reaction center. One of the remarkable features of photosynthesis, which has also attracted the attention of the physics community, is its high energy transfer efficiency during this stage [1]. There have been various theoretical and experimental attempts to understand the underlying physical mechanism responsible for such an efficient energy transfer in the biological light harvesting systems. Specifically, the role of quantum effects in excitation energy transfer has been a topic of active debate within the scientific community.

A significant body of scientific work investigates the possibility of quantum coherence at the physiological conditions and link between the coherence in light harvesting complexes and their energy transfer efficiency [2,3,4,5,6,7,8,9,10,11,12,13,14,15,16,17,18,19,20,21,22,23,24,25,26]. In recent years, several theoretical and experimental works have been published, which dispute the significance of the coherent energy transport in biological light harvesting complexes [27,28,29,30,31]. In the light of these developments, environmental noise tunes the spectral properties of the biological light harvesting complex, which can be thought of as an enhanced thermalization device, in a way to boost its transport efficiency. Recently, an experimental verification of the constructive role of the environmental noise on the energy transport properties of light harvesting complexes, based on superconducting qubits, is demonstrated by Potočnik et al. [32]. In this work, the energy transport properties of a network of three superconducting qubits, which simulate the chromophores chlorophyll molecules, and is studied upon excitation with both coherent and incoherent light. It is found that introducing noise to system can have a positive effect on the energy transmission.

These studies are of great importance because understanding the physical mechanism responsible for biological photosynthetic light harvesting systems and their limitations might lead the way in order to design biomimetic devices with comparable or even greater efficiency than their biological counterparts. Bio-inspired designs based on photosynthetic complexes have proven to be promising and efficient and are currently a subject of an active and diverse research effort encompassing quantum optics [33,34], quantum thermodynamics [35,36,37], and energy transport and conversion [38,39,40,41].

In the past decade, quantum collision models have gained popularity in order to model open quantum system dynamics [42,43,44,45,46,47,48,49,50,51]. In the collision model framework, the bath elements are modeled as a collection of acillae with which the subject system interacs, consecutively. Collision models can simulate any Markovian open system dynamics [52] but they are also particularly insightful for studies on non-Markovian open system dynamics and open quantum systems with correlations between the system and environment. Recently, there have been attempts to utilize the repeated interaction scheme to study the transport phenomena in light harvesting complexes. Chisholm et al., using a stochastic collision model, studied the transport phenomena in a quantum network model of the Fenna-Matthews-Olson (FMO) complex, and investigated the Markovian and non-Markovian effects [53]. FMO complex is a pigment-protein complex found in green sulfur bacteria and is responsible for funneling the exciton energy collected by the chlorosome antenna to the reaction center.Gallina et al. have simulated the dephasing assisted transport of a four-site network based on a collision model using the IBM Qiskit QASM simulator [54].

Inspired by the experimental setup given in [32] for simulating the energy transport in biological light harvesting complexes using superconducting qubits, we use a quantum collision model to investigate the heat transport in a similar network of qubits, whose role is to transfer energy from a hot bath to a cold bath. The goal is to study the effect of interaction between the system and a hierarchically structured environment (HSE), on the steady state heat flux. Collision models have proven to be effective in the thermodynamic analysis of a network of qubits between two thermal baths [55], a qubit coupled to a structured environment [56] and a system of qubits interacting with a thermal and a coherent thermal bath [57]. Moreover, coupling with an HSE is demonstrated to have a profound effect on coherent exciton transfer [58]. However, to the best of our knowledge, the modulation of heat flow between two thermal baths using an auxiliary HSE has not been investigated. Such an auxiliary HSE in an energy transport system conceptually resembles a system of molecules in a biological light harvesting complex used for excitonic energy transfer with the background matrix molecules supporting it. Our work utilizes the concepts and methods of the mentioned works to explore the thermodynamic behavior of an open quantum system in a complex environment.

The remainder of this work is organized as follows. In Section 2, we describe the components of our model, namely the system, the HSE, the heat baths and their interaction. Next, we describe the repeated interaction framework and quantify the heat flux. In Section 3, first, we demonstrate the effect of the system-environment and inter-environment interactions on the enhancement of the transient and steady-state heat flux and provide with a possible explanation for this behavior. We then continue to analyze the effect of coherence in the HSE on the steady-state heat flux. Finally, we discuss our results in Section 4.

## 2. The Collision Model

We consider a system made out of three qubits, namely $S_{1}$, $S_{2}$ and $S_{3}$, arranged on a line. $S_{1}$ and $S_{3}$ interact with a set of ancillary qubits representing the hot and cold thermal baths (temperatures $T_{h}$ and $T_{c}$), respectively, which we refer as $R_{n}^{h}$ and $R_{n}^{c}$. Each system qubit interacts with its nearest neighbor, In addition, we have a HSE, which is composed of a qubit *A*, with which $S_{2}$ directly interacts, and a set of ancillary qubits $B_{n}$ interacting with qubit *A*. Our main goal is to study the effect of the HSE interacting locally with $S_{2}$, and also the effect of the coherence in $B_{n}$ qubits, on energy transfer from the hot bath to the cold bath. We setup our model in the described fashion in order to mimic a similar model considered in [32], which presents an experimental investigation of the light harvesting complexes. A schematic representation of our model is given in Figure 1.

The initial states of the hot bath qubits are taken to be the thermal state of their Hamiltonian at temperature $T_{h}$, while the initial states of the system qubits, qubit *A* and the cold bath qubits are assumed to be the thermal states of their corresponding free Hamiltonian at temperature $T_{c}$. The initial state for qubits $B_{n}$ is taken to be
(1)$$\rho_{B}=p|\psi \rangle \langle \psi |+\left(1-p\right)\rho_{B}^{th}$$
where p∈[0,1] and $\rho_{B}^{th}$ is the thermal state of the qubit at temperature $T_{c}$. The state ∣ψ〉 is defined as
(2)$$|\psi \rangle =\frac{1}{Z_{e}}\left(e^{-\frac{1}{4}\omega_{e}\beta_{c}}|0\rangle +e^{\frac{1}{4}\omega_{e}\beta_{c}}|1\rangle \right),$$
where $Z_{e}=\mathrm{Tr}\left[exp\left(-\beta_{c}{\hat{H}}_{e}^{0}\right)\right]$ is the partition function and $\beta_{c}$ is the inverse temperature. The form of density matrix given by Equation (Equation 1) assures that the diagonal elements of $\rho_{B}$ and $\rho_{B}^{th}$ are identical but the diagonals are non-zero for p≠0. Hence, *p* is the parameter which controls the amount of coherence in the HSE. The free Hamiltonians for the qubits are all of the form ${\hat{H}}_{i}^{0}=\omega_{i}/2{\hat{\sigma }}_{z}^{i}$ for i=1,2,3,h,c,e in which $\omega_{i}$ and ${\hat{\sigma }}_{z}^{i}$ are the corresponding transition frequencies and Pauli *z* operators. The system Hamiltonian is given by the sum of free and interaction Hamiltonians of the system qubits as
(3)$${\hat{H}}_{\mathrm{sys}}=H_{\mathrm{sys}}^{0}+H_{\mathrm{sys}}^{I}=\sum_{i}\frac{\omega_{i}}{2}{\hat{\sigma }}_{z}^{i}+\sum_{i<j}g_{ij}\left({\hat{\sigma }}_{+}^{i}{\hat{\sigma }}_{-}^{j}+{\hat{\sigma }}_{-}^{i}{\hat{\sigma }}_{+}^{j}\right),$$
in which ${\hat{\sigma }}_{+}$ and ${\hat{\sigma }}_{-}$ are the raising and lowering operators. We model the the interaction Hamiltonian between the system and both thermal baths as an energy preserving dipole–dipole interaction
(4)$$\begin{array}{cc} {\hat{H}}_{h}^{I}= & g_{h}\left({\hat{\sigma }}_{+}^{1}{\hat{\sigma }}_{-}^{h}+{\hat{\sigma }}_{-}^{1}{\hat{\sigma }}_{+}^{h}\right) \end{array}$$
(5)$$\begin{array}{cc} {\hat{H}}_{c}^{I}= & g_{c}\left({\hat{\sigma }}_{+}^{3}{\hat{\sigma }}_{-}^{c}+{\hat{\sigma }}_{-}^{3}{\hat{\sigma }}_{+}^{c}\right). \end{array}$$
The interaction between qubits *A* and $S_{2}$ is described by a purely dephasing interaction given in the following form
(6)$${\hat{H}}_{SA}^{I}=g_{a}{\hat{\sigma }}_{z}^{2}{\hat{\sigma }}_{z}^{A}.$$
We particularly consider a different interaction Hamiltonian at this stage. The reason behind this is to make sure that we do not inject coherence into the system qubits. As stated earlier, our aim is to investigate the effect of the coherence contained in the HSE. Finally, the interaction Hamiltoinan between the qubits that belong in HSE is also chosen to be a dipole–dipole interaction
(7)$${\hat{H}}_{AB}^{I}=g_{b}\left({\hat{\sigma }}_{+}^{A}{\hat{\sigma }}_{-}^{B}+{\hat{\sigma }}_{-}^{B}{\hat{\sigma }}_{+}^{A}\right)$$

Considering all the interaction times of collisions to be equal to $t_{c}=0.01$, using $U_{i}=exp\left(-iH_{i}^{I}t_{c}/\hbar \right)$ we can calculate the corresponding time evolution operator for each Hamiltonian interaction. As a standard procedure in collision models, each collision is described by brief unitary couplings between the elements of the model. Therefore, we consider $gt_{c}$ to be smaller than 1. The repeated interaction scheme we considered is made out of the following steps:$R_{n}^{h}$ interacts with $S_{1}$$S_{1}$ interacts with $S_{2}$$S_{2}$ interacts with *A**A* interacts with $B_{n}$$S_{2}$ interacts with $S_{3}$$S_{3}$ interacts with $R_{n}^{c}$

After each set of collisions, the qubits for the hot bath $R_{n}^{h}$, cold bath $R_{n}^{c}$ and $B_{n}$ are traced out and are replaced with fresh ones, $R_{n+1}^{h}$, $R_{n+1}^{c}$ and $B_{n+1}$, in their corresponding initial states. We emphasize that qubit *A* is not reset to its initial state; it plays a role as an interface between a larger background ($B_{n}$ qubits) and the qubits forming the energy transport medium ($S_{1}$, $S_{2}$, $S_{3}$). It is important to note that the qubits *A* and $B_{n}$ qubits form an HSE to the transport medium qubits. Our collision model reflects a hierarchical coupling between the transport system and this auxiliary environment. Based on this scheme, the thermal energy transferred to the cold bath from the system at each collision step *n* can be written as
(8)$$\Delta Q_{n}=Tr\left[{\hat{H}}_{c}\left({\tilde{\rho }}_{c}^{n}-\rho_{c}^{n}\right)\right],$$
in which $\rho_{c}^{n}$ and ${\tilde{\rho }}_{c}^{n}$ are the density matrices for the cold bath qubit before and after the collision, respectively. Based on this definition, the heat current can be defined as
(9)$$J_{n}:=\frac{\Delta Q_{n}}{t_{c}}=\frac{1}{t_{c}}Tr\left[{\hat{H}}_{c}\left({\tilde{\rho }}_{c}^{n}-\rho_{c}^{n}\right)\right].$$
These sets of collisions repeat until the energy transport per collision reaches a steady state. Hence, the steady state heat flux becomes
(10)$$J_{ss}=\lim_{n\to \infty }\frac{1}{t_{c}}Tr\left[{\hat{H}}_{c}\left({\tilde{\rho }}_{c}^{n}-\rho_{c}^{n}\right)\right].$$

## 3. Results

In this section, we study the effects of $g_{a}$, $g_{b}$ and *p*, on the heat flux transported to the cold bath. We assume each qubit has the same transition frequency $\omega_{0}$ and use as our time-energy scaling parameter so that we use scaled and dimensionless parameters, where $\omega_{0}=1$. In all our calculations, the system–bath interaction couplings are taken to be $g_{c}=g_{h}=7.5$ and the collision time is $t_{c}=0.01$.

### 3.1. Effect of the System-HSE and Inter-HSE Couplings

Using the framework laid out in Section 2 we can quantify the amount of energy transferred from the hot bath to the cold bath with and without the system–HSE interaction. For the network of qubits demonstrated in Figure 1, the results are shown in Figure 2.

It is evident from Figure 2 that coupling with HSE enhances transient and steady-state heat transfer in both of the coupling regimes that is considered. A qualitative reason for one of the mechanisms responsible for this behavior can be given based on the energy level diagram of the system Hamiltonian, as given in Figure 3.

From Figure 3, we can observe that the interaction between system and the HSE modulates the energy levels of the system. The changes in the energy levels decreases the transition energy between certain eigenstates, which, therefore, makes these transitions more accessible to the hot thermal bath. As a result, the net thermal energy transfer increases upon interacting with the HSE. It is worth mentioning that due to the form of the interaction Hamiltonian $H_{SA}^{I}$ given in Equation (Equation 6), there is no energy transfer between qubits *A* and $S_{2}$, and the system qubits only transport the energy between the hot and the cold bath. In Figure 4 the steady-state heat flux as a function of $g_{a}$ and $g_{b}$ is shown when the $B_{n}$ qubits are fully thermal.

An increase in $g_{a}$ and $g_{b}$ overall, results in an enhancement of $J_{ss}$. However, for a fixed $g_{a}$ ($g_{b}$), this enhancement is more pronounced for larger values of $g_{a}$ ($g_{b}$). Therefore, we see that coupling with the HSE has a positive effect on the steady-state heat current, regardless of the initial coherence of the HSE qubits.

### 3.2. Effect of Coherence within the HSE

Presently, we turn our attention to the effect of coherence within HSE on the steady-state heat transfer. For this aim, in Figure 5, we present the dependence of $J_{ss}$ to different values of *p* and $g_{a}$.

The results indicate that there is a negligible change in $J_{ss}$ for increased values of coherence, especially for a smaller $g_{a}$. However, for larger values of $g_{a}$, it is clear that there is a noteworthy enhancement of $J_{ss}$ for a larger $g_{b}$, as indicated in Figure 5b, and a considerable non-monotonic change in $J_{ss}$ for a smaller $g_{b}$, as shown in Figure 5a.

In Figure 6, the values of $J_{ss}$ by changes in the coherence parameter *p* and $g_{b}$ is shown. There is a limited gain in $J_{ss}$ for an increased *p*, especially for smaller values of $g_{b}$. This gain is more significant for larger values of $g_{b}$. As the comparison between Figure 6a,b indicates, this effect is more pronounced for a larger $g_{a}$ coupling.

## 4. Conclusions and Discussion

Inspired by the recent developments concerning the constructive effect of noise in energy transport in biological light harvesting complexes, we have studied the effect of an auxiliary, HSE interacting with a linear system of three qubits on heat transport from a hot bath to the cold bath, connected by this system. We saw that such a coupling modulates, and in a broad range of parameters, enhances, the transient and steady state heat transport. This is partly due to the changes in the energy levels of the total system Hamiltonian, including the interaction with the HSE. Due to these changes, the thermal baths can access and induce transitions in the energy levels of the system more easily, hence enhancing the heat transport.

We conducted a full scale numerical characterization of the steady-state heat flux, with the independent parameters being the system-HSE coupling, inter-HSE coupling and coherence of the HSE qubits. Our results indicate that, for the parameter regime considered in this study, an increase in system-HSE and inter-HSE couplings corresponds to an increase in the steady-state heat current, especially for the larger values of the coupling constants. Coherence within the HSE generally has a smaller effect on the steady-state heat flux and its effect is only noticeable within a smaller parameter regime. The enhancement of the energy transfer due to the HSE coherence is more pronounced for the larger coupling constants $g_{a}$ and $g_{b}$ considered in this study.

As a future direction, one might consider the effect of non-Markovianity within the heat baths or the HSE on the heat flux by considering long range interactions between their corresponding qubits. It might also be worthwhile to study the effect of system-environment correlations or the correlations in the bath on heat transfer. Finally, our work in a sense can be understood as an open-loop control of the thermal energy through a simple network by an auxiliary environment. One might consider a more complex network of qubits and study where and how strong the local interactions with the HSE should be in order to optimally modulate the heat flux.

## Figures and Tables

**Figure 1 entropy-24-01162-f001:**
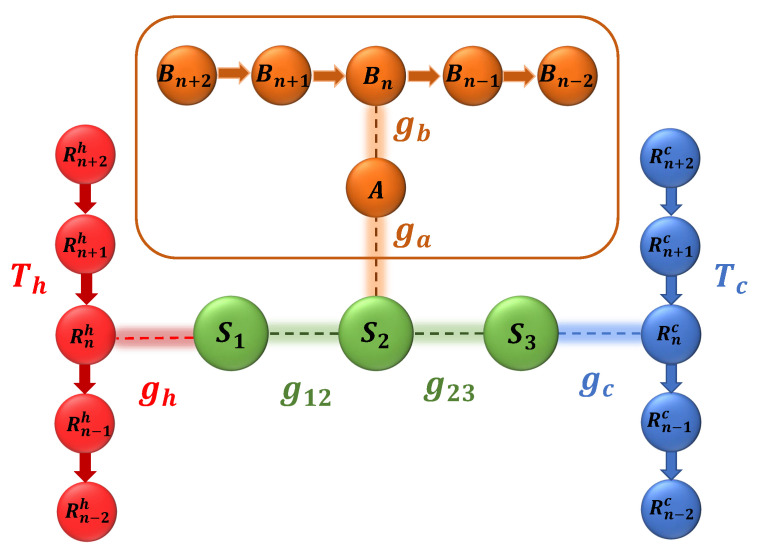
A schematic representation of the model. The system qubits, denoted by $S_{1}$, $S_{2}$ and $S_{3}$, are interacting with each other and the qubits from the cold bath, hot bath and the hierarchical environment.

**Figure 2 entropy-24-01162-f002:**
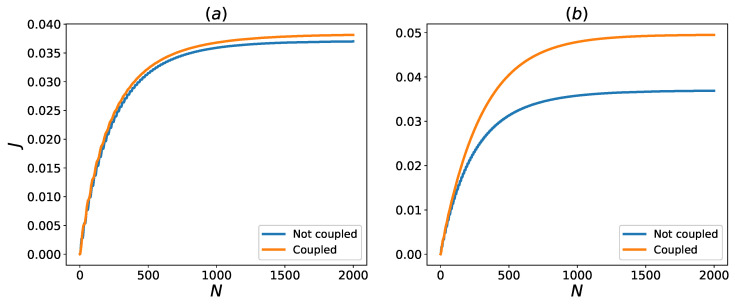
Heat flux to the cold bath, *J*, with and without interaction with the HSE for different coupling constants. The HSE are assumed to be in thermal state (p=0) and for blue curves we have $g_{a}=0$. The remaining couplings are (**a**) $g_{12}=30$, $g_{23}=15$ and for the orange curve $g_{a}=20$, $g_{b}=40$ (**b**) $g_{12}=50$, $g_{23}=25$ and for the orange curve $g_{a}=40$, $g_{b}=30$.

**Figure 3 entropy-24-01162-f003:**
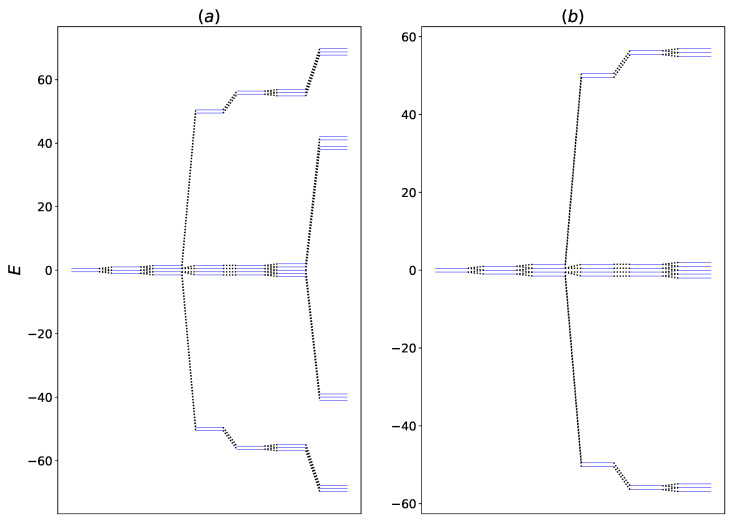
Energy level diagram for the system with and without interaction with the HSE. The interaction couplings of the system are $g_{12}=50$ and $g_{23}=20$. The system-HSE interaction is (**a**) $g_{a}=40$ (**b**) $g_{a}=0$.

**Figure 4 entropy-24-01162-f004:**
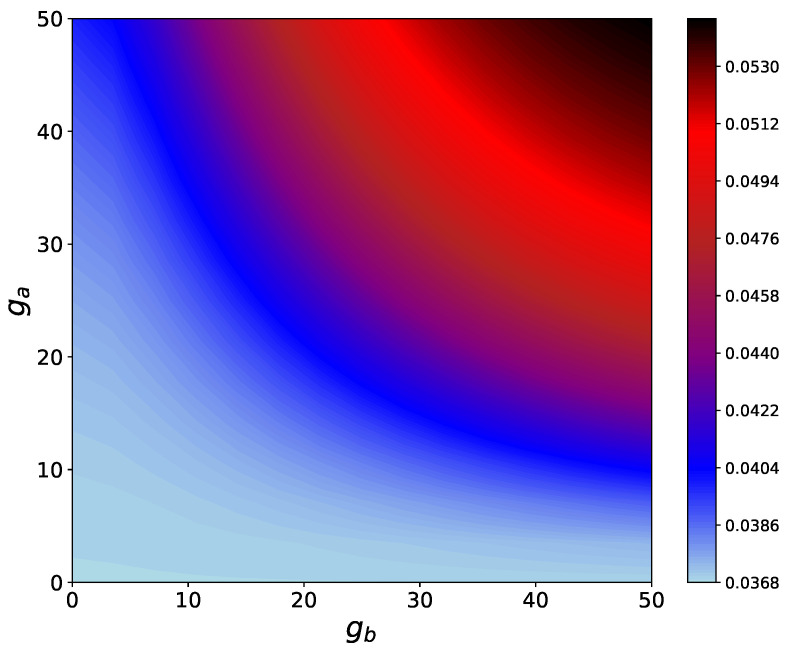
Contour plots for the steady state heat flux $J_{ss}$ vs. system-HSE and inter-HSE couplings $g_{a}$ and $g_{b}$ considering no coherence in HSE (p=0). The inter-system couplings are $g_{12}=50$ and $g_{23}=25$.

**Figure 5 entropy-24-01162-f005:**
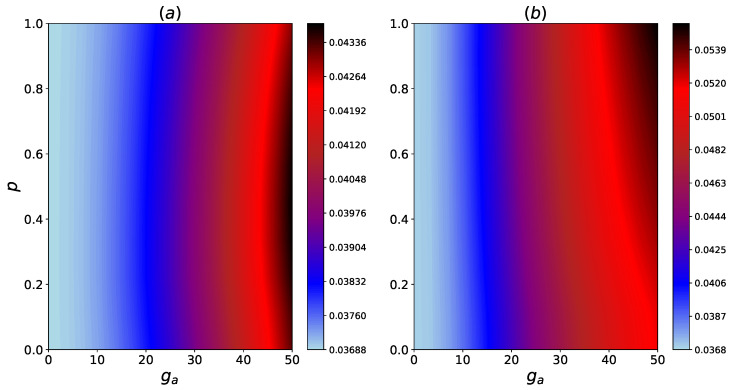
Contour plots for the steady state heat flux, $J_{ss}$ vs. system-HSE coupling, $g_{a}$, and coherence in the HSE, *p*. The couplings are (**a**) $g_{12}=50$, $g_{23}=25$, $g_{b}=10$ (**b**) $g_{12}=50$, $g_{23}=25$, $g_{b}=30$.

**Figure 6 entropy-24-01162-f006:**
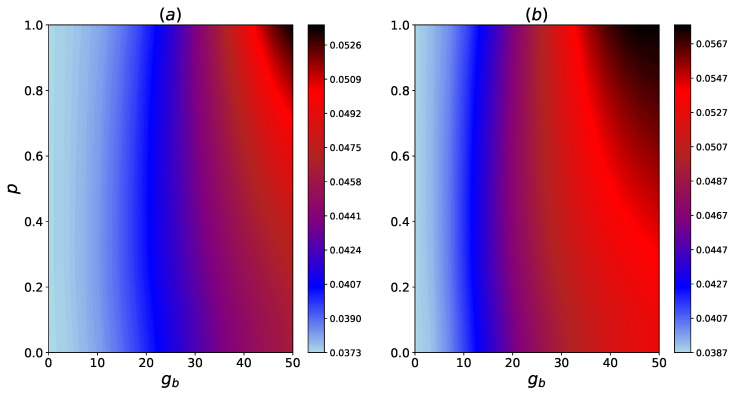
Contour plots for the steady state heat flux, $J_{ss}$, vs inter-HSE coupling, $g_{b}$, and coherence in the HSE, *p*. The couplings are (**a**) $g_{12}=50$, $g_{23}=25$, $g_{a}=20$ (**b**) $g_{12}=50$, $g_{23}=25$, $g_{a}=40$.

## Data Availability

Not applicable.

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
