# Peer review of "Environment-Assisted Modulation of Heat Flux in a Bio-Inspired System Based on Collision Model"

_entropy, 2022, doi:10.3390/e24081162_

Round 1
Reviewer 1 Report
The present paper shows a model of enhanced transfer of thermal energy between a hot and a cold bath via a series of qubits in contact with environment. It appears that the coupling with the environment enhances the heat transfer. The result is interesting and I think the paper can be published. I would recommend some more explanations on the relevance of the so many subsystems considered in Fig.1: what is the physical interpretation or physical meaning of each one, are they all relevant, what do they represent in a real system?
Reviewer 2 Report
In ‘’Environment-assisted modulation of heat flux in a bio-inspired system based on collision model’’, the authors use collision model (CM) to analyze the energy transport in a chain of thee qubits placed between two thermal baths with different temperatures, and also coupled to a hierarchically structured environment (HSE) having a coherent component. The paper aims to determine the role of the HSE in boosting the energy transfer.
The authors claim that the manuscript is the first CM-based study of this setting (qubits’ chain + 2thermal baths + HSE with coherent component); (ii) that the system proposed is mimicking a biological light harvesting process.
The paper’s main results are numerical plots showing the flux of energy transferred from the cold to the hot thermal bath as a function of the coupling amplitudes between the qubits, between the qubits and the HSE, and between the parts of the HSE. The impact of the HSE coherence is also explored in the two last panels of the paper.
The introduction of the paper is a very long overview of the most relevant physics papers on biological light harvesting processes. Until line 101, the mentioned works are not connected at all or just weekly connected to the present study. The English language needs major editing. Some statements about the existing literature are given without references, see for example lines 77-79 (if there is a significant body of work on the topic than why not to put any reference?). Despite from the length of the introduction, one of the main claims of the paper, claim (ii), is just quickly justified in line 147 without any reference or demonstrations.
The collision model treatment of the system is missing very relevant details. 1) The coupling amplitude between system and bath’s units in a CM go as t_c^{-1/2}. Is it the case here? The numerical values of the coupling amplitudes gc, gh, gb include t_c^{-1/2}? I guess so since their values are 20 times larger than the bare qubit’s frequency…2) The second order expansion of the collisional unitary evolution operator and the master equations used to produce the plots should be given explicitly. 3) The coupling between the second qubit and the qubit A is dispersive but the two qubits are resonant. How can such a coupling arise in a real physical system?
Round 2
Reviewer 1 Report
The revised manuscript can be accepted for publication.
Author Response
We would like to thank the referee for the positive response on our manuscript and all the comments and suggestions.
Reviewer 2 Report
There is a typo in line 6 of the Abstract sunthetic-> synthetic
In line 26-28 the FMO is described and appears disconnected from the results presented in the manuscript. The connection of FMO with the study of A. Pedram et al. becomes clear only in line 93, so I would put its description there.
In lines 30-36 different processes are named, they are not linked with the results presented in the manuscript but merely listed. It is not even clear if they are genuinely different mechanism or synonyms as the sentence begins with ''one of''. Either the connection with the presented results is clearly explained or this list of acronyms should be removed.
Similarly, all the papers described in lines 39-62 are not clearly connected with the results presented in the manuscript. The only connection appears to be their aim to explore the role of coherence. If this is the case then such a detailed list is useless and distracting for the reader, the sole sentence in lines 37-39 would be enough in my opinion.
Line 109 what does it mean a system of energy transfer molecules? What is an energy transfer molecule?
The sentence stating in line 128 should be reformulated as the English formulation is unclear.
Line 208 typo, within with-> with
